# A Spectral View of Adversarially Robust Features

**Shivam Garg**     **Vatsal Sharan**∗     **Brian Hu Zhang**∗     **Gregory Valiant**

**Stanford University**
Stanford, CA 94305
{shivamgarg, vsharan, bhz, gvaliant}@stanford.edu

## Abstract

Given the apparent difficulty of learning models that are robust to adversarial perturbations, we propose tackling the simpler problem of developing *adversarially robust features*. Specifically, given a dataset and metric of interest, the goal is to return a function (or multiple functions) that 1) is robust to adversarial perturbations, and 2) has significant variation across the datapoints. We establish strong connections between adversarially robust features and a natural spectral property of the geometry of the dataset and metric of interest. This connection can be leveraged to provide both robust features, and a lower bound on the robustness of *any* function that has significant variance across the dataset. Finally, we provide empirical evidence that the adversarially robust features given by this spectral approach can be fruitfully leveraged to *learn* a robust (and accurate) model.

## 1   Introduction

While machine learning models have achieved spectacular performance in many settings, including human-level accuracy for a variety of image recognition tasks, these models exhibit a striking vulnerability to *adversarial examples*. For nearly every input datapoint—including training data—a small perturbation can be carefully chosen to make the model misclassify this perturbed point. Often, these perturbations are so minute that they are not discernible to the human eye.

Since the initial work of Szegedy et al. [2013] and Goodfellow et al. [2014] identified this surprising brittleness of many models trained over high-dimensional data, there has been a growing appreciation for the importance of understanding this vulnerability. From a conceptual standpoint, this lack of robustness seems to be one of the most significant differences between humans' classification abilities (particularly for image recognition tasks), and computer models. Indeed this vulnerability is touted as evidence that computer models are not *really* learning, and are simply assembling a number of cheap and effective, but easily fooled, tricks. Fueled by a recent line of work demonstrating that adversarial examples can actually be created in the real world (as opposed to requiring the ability to edit the individual pixels in an input image) [Evtimov et al., 2017, Brown et al., 2017, Kurakin et al., 2016, Athalye and Sutskever, 2017], there has been a significant effort to examine adversarial examples from a security perspective. In certain settings where trained machine learning systems make critically important decisions, developing models that are robust to adversarial examples might be a requisite for deployment.

Despite the intense recent interest in both computing adversarial examples and on developing learning algorithms that yield robust models, we seem to have more questions than answers. In general, ensuring that models trained on high-dimensional data are robust to adversarial examples seems to be extremely difficult: for example, Athalye et al. [2018] claims to have broken six attempted defenses

---

∗Equal contribution

submitted to ICLR 2018 before the conference even happened. Additionally, we currently lack answers to many of the most basic questions concerning why adversarial examples are so difficult to avoid. What are the tradeoffs between the amount of data available, accuracy of the trained model, and vulnerability to adversarial examples? What properties of the geometry of a dataset determine whether a robust and accurate model exists?

The goal of this work is to provide a new perspective on robustness to adversarial examples, by investigating the simpler objective of finding adversarially robust *features*. Rather than trying to learn a robust function that also achieves high classification accuracy, we consider the problem of learning *any* function that is robust to adversarial perturbations with respect to *any* specified metric. Specifically, given a dataset of $d$-dimensional points and a metric of interest, can we learn *features*—namely functions from $\mathbb{R}^d \to \mathbb{R}$—which 1) are robust to adversarial perturbations of a bounded magnitude with respect to the specified metric, and 2) have significant variation across the datapoints (which precludes the trivially robust constant function).

There are several motivations for considering this problem of finding robust features: First, given the apparent difficulty of learning adversarially robust models, this is a natural first step that might help disentangle the confounding challenges of achieving robustness and achieving good classification performance. Second, given robust features, one can hope to get a robust model if the classifier used on top of these features is reasonably Lipschitz. While there are no *a priori* guarantees that the features contain any information about the labels, as we empirically demonstrate, these features seem to contain sufficient information about the geometry of the dataset to yield accurate models. In this sense, computing robust features can be viewed as a possible intermediate step in learning robust models, which might also significantly reduce the computational expense of training robust models directly. Finally, considering this simpler question of understanding robust features might yield important insights into the geometry of datasets, and the specific metrics under which the robustness is being considered (e.g. the geometry of the data under the $\ell_\infty$, or $\ell_2$, metric.) For example, by providing a lower bound on the robustness of any function (that has variance one across the datapoints), we trivially obtain a lower bound on the robustness of any classification model.

## 1.1 Robustness to Adversarial Perturbations

Before proceeding, it will be useful to formalize the notion of robustness (or lack thereof) to adversarial examples. The following definition provides one natural such definition, and is given in terms of a distribution $D$ from which examples are drawn, and a specific metric, $dist(\cdot, \cdot)$ in terms of which the magnitude of perturbations will be measured.

**Definition 1.** *A function $f : \mathbb{R}^d \to \mathbb{R}$ is said to be $(\varepsilon, \delta, \gamma)$ robust to adversarial perturbations for a distribution $D$ over $\mathbb{R}^d$ with respect to a distance metric $dist : \mathbb{R}^d \times \mathbb{R}^d \to \mathbb{R}$ if, for a point $x$ drawn according to $D$, the probability that there exists $x'$ such that $dist(x, x') \leq \varepsilon$ and $|f(x) - f(x')| \geq \delta$, is bounded by $\gamma$. Formally,*

$$\Pr_{x \sim D}[\exists x' \ s.t. \ dist(x, x') \leq \varepsilon \ and \ |f(x) - f(x')| \geq \delta] \leq \gamma.$$

In the case that the function $f$ is a binary classifier, if $f$ is $(\varepsilon, 1, \gamma)$ robust with respect to the distribution $D$ of examples and a distance metric $d$, then even if adversarial perturbations of magnitude $\varepsilon$ are allowed, the classification accuracy of $f$ can suffer by at most $\gamma$.

Our approach will be easier to describe, and more intuitive, when viewed as a method for assigning feature values to an entire dataset. Here the goal is to map each datapoint to a feature value (or set of values), which is robust to perturbations of the points in the dataset. Given a dataset $X$ consisting of $n$ points in $\mathbb{R}^d$, we desire a function $F$ that takes as input $X$, and outputs a vector $F(X) \in \mathbb{R}^n$; such a function $F$ is robust for a dataset $X$ if, for all $X'$ obtained by perturbing points in $X$, $F(X)$ and $F(X')$ are close.

Formally, let $\mathcal{X}$ be the set of all datasets consisting of $n$ points in $\mathbb{R}^d$, and $\|\cdot\|$ denote the $\ell_2$ norm. For notational convenience, we will use $F_X$ and $F(X)$ interchangeably, and use $F_X(x)$ to denote the feature value $F$ associates with a point $x \in X$. We overload $dist(\cdot, \cdot)$ to define distance between two ordered sets $X = (x_1, x_2, \ldots, x_n)$ and $X' = (x'_1, x'_2, \ldots, x'_n)$ as $dist(X, X') = max_{i \in [n]} dist(x_i, x'_i)$. With these notations in place, we define a robust function as follows:

**Definition 2.** *A function $F : \mathcal{X} \to \mathbb{R}^n$ is said to be $(\varepsilon, \delta)$ robust to adversarial perturbations for a dataset $X$ with respect to a distance metric $dist(\cdot, \cdot)$ as defined above, if, for all datasets $X'$ such that $dist(X, X') \leq \epsilon$, $\|F(X) - F(X')\| \leq \delta$.*

If a function $F$ is $(\epsilon, \delta)$ robust for a dataset $X$, it implies that feature values of $99\%$ of the points in $X$ will not vary by more than $\frac{10\delta}{\sqrt{n}}$ if we were to perturb all points in $X$ by at most $\epsilon$.

As in the case of robust functions of single datapoints, to preclude the possibility of some trivial functions we require $F$ to satisfy certain conditions: 1) $F_X$ should have significant variance across points, say, $\sum_i \left( F_X(x_i) - \mathbb{E}_{x \sim Unif(X)}[F_X(x)] \right)^2 = 1$. 2) Changing the order of points in dataset X should not change $F_X$, that is, for any permutation $\sigma : \mathbb{R}^n \to \mathbb{R}^n$, $F_{\sigma(X)} = \sigma(F_X)$. Given a data distribution $D$, and a threshold $\epsilon$, the goal will be to find a function $F$ that is as robust as possible, in expectation, for a dataset $X$ drawn from $D$.

We mainly follow Definition 2 throughout the paper as the ideas behind our proposed features follow more naturally under that definition. However, we briefly discuss how to extend these ideas to come up with robust features of single datapoints (Definition 1) in section 2.1.

## 1.2 Summary of Results

In Section 2, we describe an approach to constructing features using spectral properties of an appropriately defined graph associated with a dataset in question. We show provable bounds for the adversarial robustness of these features. We also show a synthetic setting in which some of the existing models such as neural networks, and nearest-neighbor classifier are known to be vulnerable to adversarial perturbations, while our approach provably works well. In Section 3, we show a lower bound which, in certain parameter regimes, implies that if our spectral features are not robust, then no robust features exist. The lower bound suggests a fundamental connection between the spectral properties of the graph obtained from the dataset, and the inherent extent to which the data supports adversarial robustness. To explore this connection further, in Section 5, we show empirically that spectral properties do correlate with adversarial robustness. In Section 5, we also test our adversarial features on the downstream task of classification on adversarial images, and obtain positive results. Due to space constraints, we have deferred all the proofs to the supplementary material.

## 1.3 Shortcomings and Future Work

Our theory and empirics indicate that there may be fundamental connections between spectral properties of graphs associated with data and the inherent robustness to adversarial examples. A worthwhile future direction is to further clarify this connection, as it may prove illuminating and fruitful. Looking at the easier problem of finding adversarial features also presents the opportunity of developing interesting sample-complexity results for security against adversarial attacks. Such results may be much more difficult to prove for the problem of adversarially robust classification, since generalization is not well understood (even in the non-adversarial setting) for classification models such as neural networks.

Our current approach involves computing distances between all pair of points, and performing an eigenvector computation on a Laplacian matrix of a graph generated using these distances. Both of these steps are computationally expensive operations, and future work could address improving the efficiency of our approach. In particular, it seems likely that similar spectral features can be approximated without computing all the pairwise distances, which would result in significant speed-up. We also note that our experiments for testing our features on downstream classification tasks on adversarial data is based on transfer attacks, and it may be possible to degrade this performance using stronger attacks. The main takeaway from this experiment is that our conceptually simple features along with a linear classifier is able to give competitive results for reasonable strong attacks. Future works can possibly explore using robustly trained models on top of these spectral features, or using a spectral approach to distill the middle layers of neural networks.

## 1.4 Related Work

One of the very first methods proposed to defend against adversarial examples was adversarial training using the *fast gradient sign method* (FGSM) [Goodfellow et al., 2014], which involves taking

a step in the direction of the gradient of loss with respect to data, to generate adversarial examples, and training models on these examples. Later, Madry et al. [2017] proposed a stronger *projected gradient descent* (PGD) training which essentially involves taking multiple steps in the direction of the gradient to generate adversarial examples followed by training on these examples. More recently, Kolter and Wong [2017], Raghunathan et al. [2018], and Sinha et al. [2017] have also made progress towards training provably adversarially robust models. There have also been efforts towards proving lower bounds on the adversarial accuracy of neural networks, and using these lower bounds to train robust models [Hein and Andriushchenko, 2017, Peck et al., 2017]. Most prior work addresses the question of how to fix the adversarial examples problem, and there is less work on identifying why this problem occurs in the first place, or highlighting which geometric properties of datasets make them vulnerable to adversarial attacks. Two recent works specifically address the "why" question: Fawzi et al. [2018] give lower bounds on robustness given a specific generative model of the data, and Schmidt et al. [2018] and Bubeck et al. [2018] describe settings in which limited computation or data are the primary bottleneck to finding a robust classifier. In this work, by considering the simpler task of coming up with robust features, we provide a different perspective on both the questions of "why" adversarial perturbations are effective, and "how" to ensure robustness to such attacks.

### 1.5  Background: Spectral Graph Theory

Let $G = (V(G), E(G))$ be an undirected, possibly weighted graph, where for notational simplicity $V(G) = \{1, \ldots, n\}$. Let $A = (a_{ij})$ be the adjacency matrix of $G$, and $D$ be the the diagonal matrix whose $i$th diagonal entry is the sum of edge weights incident to vertex $i$. The matrix $L = D - A$ is called the *Laplacian matrix* of the graph $G$. The quadratic form, and hence the eigenvalues and eigenvectors, of $L$ carry a great deal of information about $G$. For example, for any $v \in \mathbb{R}^n$, we have

$$v^T L v = \sum_{(i,j) \in E(G)} a_{ij}(v_i - v_j)^2.$$

It is immediately apparent that $L$ has at least one eigenvalue of 0: the vector $v_1 = (1, 1, \ldots, 1)$ satisfies $v^T L v = 0$. Further, the second (unit) eigenvector is the solution to the minimization problem

$$\min_v \sum_{(i,j) \in E} a_{ij}(v_i - v_j)^2 \quad \text{s.t.} \quad \sum_i v_i = 0; \quad \sum_i v_i^2 = 1.$$

In other words, the second eigenvector assigns values to the vertices such that the average value is 0, the variance of the values across the vertices is 1, and among such assignments, minimizes the sum of the squares of the discrepancies between neighbors. For example, in the case that the graph has two (or more) connected components, this second eigenvalue is 0, and the resulting eigenvector is constant on each connected component.

Our original motivation for this work is the observation that, at least superficially, this characterization of the second eigenvector sounds similar in spirit to a characterization of a robust feature: here, neighboring vertices should have similar value, and for robust features, close points should be mapped to similar values. The crucial question then is how to formalize this connection. Specifically, is there a way to construct a graph such that the neighborhood structure of the graph captures the neighborhood of datapoints with respect to the metric in question? We outline one such construction in Section 2.

We will also consider the *normalized* or *scaled Laplacian*, which is defined by

$$L_{\text{norm}} = D^{-1/2}(D - A)D^{-1/2} = I - D^{-1/2} A D^{-1/2}.$$

The scaled Laplacian normalizes the entries of $L$ by the total edge weights incident to each vertex, so that highly-irregular graphs do not have peculiar behavior. For more background on spectral graph theory, we refer the readers to Spielman [2007] and Chung [1997].

## 2  Robust Features

In this section, we describe a construction of robust features, and prove bounds on their robustness. Let $X = (x_1, \ldots, x_n)$ be our dataset, and let $\varepsilon > 0$ be a threshold for attacks. We construct a robust feature $F_X$ using the second eigenvector of the Laplacian of a graph corresponding to $X$, defined in terms of the metric in question. Formally, given the dataset $X$, and a distance threshold parameter $T > 0$ which possibly depends on $\varepsilon$, we define $F_X$ as follows:

Define $G(X)$ to be the graph whose nodes correspond to points in $X$, i.e., $\{x_1, \ldots, x_n\}$, and for which there is an edge between nodes $x_i$ and $x_j$, if $dist(x_i, x_j) \leq T$. Let $L(X)$ be the (un-normalized) Laplacian of $G(X)$, and let $\lambda_k(X)$ and $v_k(X)$ be its $k$th smallest eigenvalue and a corresponding unit eigenvector. In all our constructions, we assume that the first eigenvector $v_1(X)$ is set to be the unit vector proportional to the all-ones vector. Now define $F_X(x_i) = v_2(X)_i$; i.e. the component of $v_2(X)$ corresponding to $x_i$. Note that $F_X$ defined this way satisfies the requirement of sufficient variance across points, namely, $\sum_i (F_X(x_i) - \mathbb{E}_{x \sim Unif(X)}[F_X(x)])^2 = 1$, since $\sum_i v_2(X)_i = 0$ and $\|v_2(X)\| = 1$.

We now give robustness bounds for this choice of feature $F_X$. To do this, we will need slightly more notation. For a fixed $\varepsilon > 0$, define the graph $G^+(X)$ to be the graph with the same nodes as $G(X)$, except that the threshold for an edge is $T + 2\varepsilon$ instead of $T$. Formally, in $G^+(X)$, there is an edge between $x_i$ and $x_j$ if $dist(x_i, x_j) \leq T + 2\epsilon$. Similarly, define $G^-(X)$ to be the graph with same set of nodes, with the threshold for an edge being $T - 2\varepsilon$. Define $L^+(X), \lambda_k^+(X), v_k^+(X), L^-(X), \lambda_k^-(X), v_k^-(X)$ analogously to the earlier definitions. In the following theorem, we give robustness bounds on the function $F$ as defined above.

**Theorem 1.** *For any pair of datasets $X$ and $X'$, such that $dist(X, X') \leq \varepsilon$, the function $F : \mathcal{X}_n \to \mathbb{R}^n$ obtained using the second eigenvector of the Laplacian as defined above satisfies*

$$\min(\|F(X) - F(X')\|, \|(-F(X)) - (F(X'))\|) \leq \left(2\sqrt{2}\right)\sqrt{\frac{\lambda_2^+(X) - \lambda_2^-(X)}{\lambda_3^-(X) - \lambda_2^-(X)}}.$$

Theorem 1 essentially guarantees that the features, as defined above, are robust up to sign-flip, as long as the eigengap between the second and third eigenvalues is large, and the second eigenvalue does not change significantly if we slightly perturb the distance threshold used to determine whether an edge exists in the graph in question. Note that flipping signs of the feature values of all points in a dataset (including training data) does not change the classification problem for most common classifiers. For instance, if there exists a linear classifier that fits points with features $F_X$ well, then a linear classifier can fit points with features $-F_X$ equally well. So, up to sign flip, the function $F$ is $(\varepsilon, \delta_X)$ robust for dataset $X$, where $\delta_X$ corresponds to the bound given in Theorem 1.

To understand this bound better, we discuss a toy example. Consider a dataset $X$ that consists of two clusters with the property that the distance between any two points in the same cluster is at most $4\epsilon$, and the distance between any two points in different clusters is at least $10\epsilon$. Graph $G(X)$ with threshold $T = 6\epsilon$, will have exactly two connected components. Note that $v_2(X)$ will perfectly separate the two connected components with $v_2(X)_i$ being $\frac{1}{\sqrt{n}}$ if $i$ belongs to component 1, and $\frac{-1}{\sqrt{n}}$ otherwise. In this simple case, we conclude immediately that $F_X$ is perfectly robust: perturbing points by $\epsilon$ cannot change the connected component any point is identified with. Indeed, this agrees with Theorem 1: $\lambda_2^+ = \lambda_2^- = 0$ since the two clusters are at a distance $> 10\varepsilon$.

Next, we briefly sketch the idea behind the proof of Theorem 1. Consider the second eigenvector $v_2(X')$ of the Laplacian of the graph $G(X')$ where dataset $X'$ is obtained by perturbing points in $X$. We argue that this eigenvector can not be too far from $v_2^-(X)$. For the sake of contradiction, consider the extreme case where $v_2(X')$ is orthogonal to $v_2^-(X)$. If the gap between the second and third eigenvalue of $G^-(X)$ is large, and the difference between $\lambda_2(X')$ and $\lambda_2^-(X)$ is small, then by replacing $v_3^-(X)$ with $v_2(X')$ as the third eigenvector of $G^-(X)$, we get a much smaller value for $\lambda_3^-(X)$, which is not possible. Hence, we show that the two eigenvectors in consideration can not be orthogonal. The proof of the theorem extends this argument to show that $v_2(X')$, and $v_2^-(X)$ need to be close if we have a large eigengap for $G^-(X)$, and a small gap between $\lambda_2(X')$ and $\lambda_2^-(X)$. Using a similar argument, one can show that $v_2(X)$ and $v_2^-(X)$, also need to be close. Applying the triangle inequality, we get that $v_2(X)$ and $v_2(X')$ are close. Also, since we do not have any control over $\lambda_2(X')$, we use an upper bound on it given by $\lambda_2(X+)$, and state our result in terms of the gap between $\lambda_2(X^+)$ and $\lambda_2(X^-)$.

The approach described above also naturally yields a construction of a *set* of robust features by considering the higher eigenvectors of Laplacian. We define the $i^{\text{th}}$ feature vector for a dataset $X$ as $F_X^i = v_{i+1}(X)$. As the eigenvectors of a symmetric matrix are orthogonal, this gives us a set of $k$ diverse feature vectors $\{F_X^1, F_X^2, \ldots, F_X^k\}$. Let $F_X(x) = (F_X^1(x), F_X^k(x), \ldots, F_X^k(x))^T$ be a

$k$-dimensional column vector denoting the feature values for point $x \in X$. In the following theorem, we give robustness bounds on these feature vectors.

**Theorem 2.** *For any pair of datasets $X$ and $X'$, such that $dist(X, X') \leq \varepsilon$, there exists a $k \times k$ invertible matrix $M$, such that the features $F_X$ and $F_{X'}$ as defined above satisfy*

$$\sqrt{\sum_{i \in [n]} \|MF_X(x_i) - F_{X'}(x_i')\|^2} \leq \left(2\sqrt{2k}\right)\sqrt{\frac{\lambda_{k+1}^+(X) - \lambda_2^-(X)}{\lambda_{k+2}^-(X) - \lambda_2^-(X)}}$$

Theorem 2 is a generalization of Theorem 1, and gives a bound on the robustness of feature vectors $F_X$ up to linear transformations. Note that applying an invertible linear transformation to all the points in a dataset (including training data) does not alter the classification problem for models invariant under linear transformations. For instance, if there exists a binary linear classifier given by vector $w$, such that $sign(w^T F_X(x))$ corresponds to the true label for point $x$, then the classifier given by $(M^{-1})^T w$ assigns the correct label to linearly transformed feature vector $MF_X(x)$.

## 2.1 Extending a Feature to New Points

In the previous section, we discussed how to get robust features for points in a dataset. In this section, we briefly describe an extension of that approach to get robust features for points outside the dataset, as in Definition 1.

Let $X = \{x_1, \ldots, x_n\} \subset \mathbb{R}^d$ be the training dataset drawn from some underlying distribution $D$ over $\mathbb{R}^d$. We use $X$ as a reference to construct a robust function $f_X : \mathbb{R}^d \to \mathbb{R}$. For the sake of convenience, we drop the subscript $X$ from $f_X$ in the case where the dataset in question is clear. Given a point $x \in \mathbb{R}^d$, and a distance threshold parameter $T > 0$, we define $f(x)$ as follows:

Define $G(X)$ and $G(x)$ to be graphs whose nodes are points in dataset $X$, and $\{x\} \cup X = \{x_0 = x, x_1, \ldots, x_n\}$ respectively, and for which there is an edge between nodes $x_i$ and $x_j$, if $dist(x_i, x_j) \leq T$. Let $L(x)$ be the Laplacian of $G(x)$, and let $\lambda_k(x)$ and $v_k(x)$ be its $k$th smallest eigenvalue and a corresponding unit eigenvector. Similarly, define $L(X)$, $\lambda_k(X)$ and $v_k(X)$ for $G(X)$. In all our constructions, we assume that the first eigenvectors $v_1(X)$ and $v_1(x)$ are set to be the unit vector proportional to the all-ones vector. Now define $f(x) = v_2(x)_0$; i.e. the component of $v_2(x)$ corresponding to $x_0 = x$. Note that the eigenvector $v_2(x)$ has to be picked "consistently" to avoid signflips in $f$ as $-v_2(x)$ is also a valid eigenvector. To resolve this, we select the eigenvector $v_2(x)$ to be the eigenvector (with eigenvalue $\lambda_2(x)$) whose last $|X|$ entries has the maximum inner product with $v_2(X)$.

We now state a robustness bound for this feature $f$ as per Definition 1. For a fixed $\varepsilon > 0$ define the graph $G^+(x)$ to be the graph with the same nodes and edges of $G(x)$, except that the threshold for $x_0 = x$ is $T + \varepsilon$ instead of $T$. Formally, in $G^+(x)$, there is an edge between $x_i$ and $x_j$ if:

    (a) $i = 0$ or $j = 0$, and $dist(x_i, x_j) \leq T + \varepsilon$; or

    (b) $i > 0$ and $j > 0$, and $dist(x_i, x_j) \leq T$.

Similarly, define $G^-(x)$ to be the same graph with $T + \varepsilon$ replaced with $T - \varepsilon$. Define $L^+, \lambda_k^+, v_k^+, L^-, \lambda_k^-, v_k^-$ analogously to the earlier definitions. In the following theorem, we give a robustness bound on the function $f$ as defined above.

**Theorem 3.** *For a sufficiently large training set size $n$, if $\mathbb{E}_{X \sim D}\left[(\lambda_3(X) - \lambda_2(X))^{-1}\right] \leq c$ for some small enough constant $c$, then with probability $0.95$ over the choice of $X$, the function $f_X : \mathbb{R}^d \to \mathbb{R}$ as defined above satisfies $\Pr_{x \sim D}[\exists x' \text{ s.t. } dist(x, x') \leq \varepsilon \text{ and } |f_X(x) - f_X(x')| \geq \delta_x] \leq 0.05$, for*

$$\delta_x = \left(6\sqrt{2}\right)\sqrt{\frac{\lambda_2^+(x) - \lambda_2^-(x)}{\lambda_3^-(x) - \lambda_2^-(x)}}.$$

*This also implies that with probability $0.95$ over the choice of $X$, $f_X$ is $(\epsilon, 20\,\mathbb{E}_{x \sim D}[\delta_x], 0.1)$ robust as per Definition 1.*

This bound is very similar to bound obtained in Theorem 1, and says that the function $f$ is robust, as long as the eigengap between the second and third eigenvalues is sufficiently large for $G(X)$ and

$G^-(x)$, and the second eigenvalue does not change significantly if we slightly perturb the distance threshold used to determine whether an edge exists in the graph in question. Similarly, one can also obtain a set of $k$ features, by taking the first $k$ eigenvectors of $G(X)$ prepended with zero, and projecting them onto the bottom-$k$ eigenspace of $G(x)$.

## 3    A Lower Bound on Adversarial Robustness

In this section, we show that spectral properties yield a lower bound on the robustness of any function on a dataset. We show that if there exists an $(\varepsilon, \delta)$ robust function $F'$ on dataset $X$, then the spectral approach (with appropriately chosen threshold), will yield an $(\varepsilon', \delta')$ robust function, where the relationship between $\varepsilon, \delta$ and $\varepsilon', \delta'$ is governed by easily computable properties of the dataset, $X$. This immediately provides a way of establishing a bound on the best possible robustness that dataset $X$ could permit for perturbations of magnitude $\varepsilon$. Furthermore it suggests that the spectral properties of the neighborhood graphs we consider, may be inherently related to the robustness that a dataset allows. We now formally state our lower bound:

**Theorem 4.** *Assume that there exists some $(\varepsilon, \delta)$ robust function $F^*$ for the dataset $X$ (not necessarily constructed via the spectral approach). For any threshold $T$, let $G_T$ be the graph obtained on $X$ by thresholding at $T$. Let $d_T$ be the maximum degree of $G_T$. Then the feature $F$ returned by the spectral approach on the graph $G_{2\varepsilon/3}$ is at least $(\varepsilon/6, \delta')$ robust (up to sign), for*

$$\delta' = \delta\sqrt{\frac{8(d_\varepsilon + 1)}{\lambda_3(G_{\varepsilon/3}) - \lambda_2(G_{\varepsilon/3})}}.$$

The bound gives reasonable guarantees when the degree is small and the spectral gap is large. To produce meaningful bounds, the neighborhood graph must have some structure at the threshold in question; in many practical settings, this would require an extremely large dataset, and hence this bound is mainly of theoretical interest at this point. Still, our experimental results in Section 5 empirically validate the hypothesis that spectral properties have implications for the robustness of any model: we show that the robustness of an adversarially trained neural network on different data distributions correlates with the spectral properties of the distribution.

## 4    Synthetic Setting: Adversarial Spheres

Gilmer et al. [2018] devise a situation in which they are able to show in theory that training adversarially robust models is difficult. The authors describe the "concentric spheres dataset", which consists of—as the name suggests—two concentric $d$-dimensional spheres, one of radius 1 and one of radius $R > 1$. The authors then argue that any classifier that misclassifies even a small fraction of the inner sphere will have a significant drop in adversarial robustness.

We argue that our method, in fact, yields a near-perfect classifier—one that makes almost no errors on natural or adversarial examples—even when trained on a modest amount of data. To see this, consider a sample of $2N$ training points from the dataset, $N$ from the inner sphere and $N$ from the outer sphere. Observe that the distance between two uniformly chosen points on a sphere of radius $r$ is close to $r\sqrt{2}$. In particular, the median distance between two such points is exactly $r\sqrt{2}$, and with high probability for large $d$, the distance will be within some small radius $\varepsilon$ of $r\sqrt{2}$. Thus, for distance threshold $\sqrt{2} + 2\varepsilon$, after adding a new test point to the training data, we will get a graph with large clique corresponding to the inner sphere, and isolated points on the outer sphere, with high probability. This structure doesn't change by perturbing the test point by $\epsilon$, resulting in a robust classifier. We now formalize this intuition.

Let the inner sphere be of radius one, and outer sphere be of some constant radius $R > 1$. Let $\varepsilon = (R - 1)/8$ be the radius of possible perturbations. Then we can state the following:

**Theorem 5.** *Pick initial distance threshold $T = \sqrt{2} + 2\varepsilon$ in the $\ell_2$ norm, and use the first $N + 1$ eigenvectors as proposed in Section 2.1 to construct a $(N + 1)$-dimensional feature map $f : \mathbb{R}^d \to \mathbb{R}^{N+1}$. Then with probability at least $1 - N^2 e^{-\Omega(d)}$ over the random choice of training set, $f$ maps the entire inner sphere to the same point, and the entire outer sphere to some other point, except for a $\gamma$-fraction of both spheres, where $\gamma = N e^{-\Omega(d)}$. In particular, $f$ is $(\varepsilon, 0, \gamma)$-robust.*

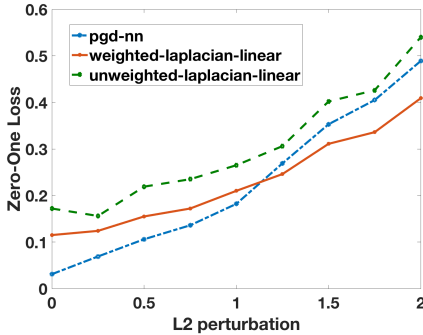

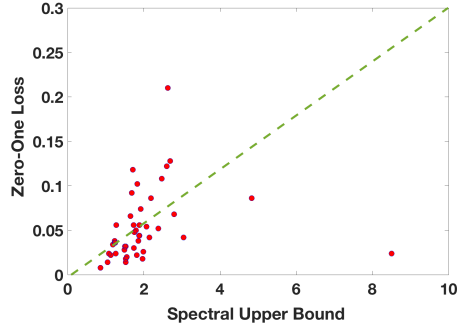

Figure 1: Comparison of performance on adversarially perturbed MNIST data .

Figure 2: Performance on adversarial data vs our upper bound.

The extremely nice form of the constructed feature $f$ in this case means that, if we use half of the training set to get the feature map $f$, and the other half to train a linear classifier (or, indeed, any nontrivial model at all) trained on top of this feature, this will yield a near-perfect classifier even against adversarial attacks. The adversarial spheres example is a case in which our method allows us to make a robust classifier, but other common methods do not. For example, nearest-neighbors will fail at classifying the outer sphere (since points on the outer sphere are generally closer to points on the inner sphere than to other points on the outer sphere), and Gilmer et al. [2018] demonstrate in practice that training adversarially robust models on the concentric spheres dataset using standard neural network architectures is extremely difficult when the dimension $d$ grows large.

# 5 Experiments

## 5.1 Image Classification: The MNIST Dataset

While the main focus of our work is to improve the conceptual understanding of adversarial robustness, we also perform experiments on the MNIST dataset. We test the efficacy of our features by evaluating them on the downstream task of classifying adversarial images. We used a subset of MNIST dataset, which is commonly used in discussions of adversarial examples [Goodfellow et al., 2014, Szegedy et al., 2013, Madry et al., 2017]. Our dataset has 11,000 images of hand written digits from zero to nine, of which 10,000 images are used for training, and rest for test. We compare three different models, the specifics of which are given below:

**Robust neural network (pgd-nn)**: We consider a fully connected neural network with one hidden layer having 200 units, with ReLU non-linearity, and cross-entropy loss. We use PyTorch implementation of Adam [Kingma and Ba, 2014] for optimization with a step size of $0.001$. To obtain a robust neural network, we generate adversarial examples using projected gradient descent for each mini-batch, and train our model on these examples. For projected gradient descent, we use a step size of $0.1$ for $40$ iterations.

**Spectral features obtained using scaled Laplacian, and linear classifier (unweighted-laplacian-linear)**: We use the $\ell_2$ norm as a distance metric, and distance threshold $T = 9$ to construct a graph on all 11,000 data points. Since the distances between training points are highly irregular, our constructed graph is also highly irregular; thus, we use the scaled Laplacian to construct our features. Our features are obtained from the 20 eigenvectors corresponding to $\lambda_2$ to $\lambda_{21}$. Thus each image is mapped to a feature vector in $\mathbb{R}^{20}$. On top of these features, we use a linear classifier with cross-entropy loss for classification. We train the linear classifier using 10,000 images, and test it on 1,000 images obtained by adversarially perturbing test images.

**Spectral features obtained using scaled Laplacian with weighted edges, and linear classifier (weighted-laplacian-linear)**: This is similar to the previous model, with the only difference being the way in which the graph is constructed. Instead of using a fixed threshold, we have weighted edges between all pairs of images, with the weight on the edge between image $i$ and $j$ being

$\exp\left(-0.1\|x_i - x_j\|_2^2\right)$. As before, we use 20 eigenvectors corresponding to the scaled Laplacian of this graph, with a linear classifier for classification.

Note that generating our features involve computing distances between all pair of images, followed by an eigenvector computation. Therefore, finding the gradient (with respect to the image coordinates) of classifiers built on top of these features is computationally extremely expensive. As previous works [Papernot et al., 2016] have shown that transfer attacks can successfully fool many different models, we use transfer attacks using adversarial images corresponding to robust neural networks (pgd-nn).

The performance of these models on adversarial data is shown in figure 1. We observe that weighted-laplacian-linear performs better than pgd-nn on large enough perturbations. Note that it is possible that robustly trained deep convolutional neural nets perform better than our model. It is also possible that the performance of our models may deteriorate with stronger attacks. Still, our conceptually simple features, with just a linear classifier on top, are able to give competitive results against reasonably strong adversaries. It is possible that training robust neural networks on top of these features, or using such features for the middle layers of neural nets may give significantly more robust models. Therefore, our experiments should be considered mainly as a proof of concept, indicating that spectral features may be a useful tool in one's toolkit for adversarial robustness.

We also observe that features from weighted graphs perform better than their unweighted counterpart. This is likely because the weighted graph contains more information about the distances, while most of this information is lost via thresholding in the unweighted graph.

### 5.2 Connection Between Spectral Properties and Robustness

We hypothesize that the spectral properties of the graph associated with a dataset has fundamental connections with its adversarial robustness. The lower bound shown in section 3 sheds some more light on this connection. In Theorem 1, we show that adversarial robustness is proportional to $\sqrt{(\lambda_2^+ - \lambda_2^-)/(\lambda_3^- - \lambda_2^-)}$. To study this connection empirically, we created 45 datasets corresponding to each pair of digits in MNIST. As we expect some pairs of digits to be less robust to adversarial perturbations than others, we compare our spectral bounds for these various datasets, to their observed adversarial accuracies.

**Setup:** The dataset for each pair of digits has 5000 data points, with 4000 points used as the training set, and 1000 points used as the test set. Similarly to the previous subsection, we trained robust neural nets on these datasets. We considered fully connected neural nets with one hidden layer having 50 units, with ReLU non-linearity, and cross-entropy loss. For each mini-batch, we generated adversarial examples using projected gradient descent with a step size of 0.2 for 20 iterations, and trained the neural net on these examples. Finally, to test this model, we generated adversarial perturbations of size 1 in $\ell_2$ norm to obtain the adversarial accuracy for all 45 datasets.

To get a bound for each dataset $X$, we generated two graphs $G^-(X)$, and $G^+(X)$ with all 5000 points (not involving adversarial data). We use the $\ell_2$ norm as a distance metric. The distance threshold $T$ for $G^-(X)$ is set to be the smallest value such that each node has degree at least one, and the threshold for $G^+(X)$ is two more than that of $G^-(X)$. We calculated the eigenvalues of the scaled Laplacians of these graphs to obtain our theoretical bounds.

**Observations:** As shown in Figure 2, we observe some correlation between our upper bounds and the empirical adversarial robustness of the datasets. Each dataset is represented by a point in Figure 2, where the x-axis is proportional to our bound, and the y-axis indicates the zero-one loss of the neural nets on adversarial examples generated from that dataset. The correlation is 0.52 after removing the right-most outlier. While this correlation is not too strong, it suggests some connection between our spectral bounds on the robustness and the empirical robustness of certain attack/defense heuristics.

## 6 Conclusion

We considered the task of learning adversarially robust features as a simplification of the more common goal of learning adversarially robust classifiers. We showed that this task has a natural connection to spectral graph theory, and that spectral properties of a graph associated to the underlying data have implications for the robustness of any feature learned on the data. We believe that exploring this simpler task of learning robust features, and further developing the connections to spectral graph theory, are promising steps towards the end goal of building robust machine learning models.

**Acknowledgments:** This work was supported by NSF awards CCF-1704417 and 1813049, and an ONR Young Investigator Award (N00014-18-1-2295).

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
