[Supplementary Material · spectral-view-adversarially-neurips-supplementary.pdf]

# A Spectral View of Adversarially Robust Features: Supplementary Material

**Shivam Garg**    **Vatsal Sharan**[*]    **Brian Hu Zhang**[*]    **Gregory Valiant**

**Stanford University**
Stanford, CA 94305
{shivamgarg, vsharan, bhz, gvaliant}@stanford.edu

## Proof of Theorems 1 and 2

Before proving the theorems, it will be useful to prove a few helper lemmas that we use at various places.

**Lemma 1.** *Let $G$ and $G^-$ be two graphs with $V(G^-) = V(G)$, $E(G^-) \subseteq E(G)$. Let $L$ and $L^-$ be their respective Laplacians. Then $v^T L^- v \leq v^T L v$ for all vectors $v$.*

*Proof.* The proof follows from the definition of Laplacian:

$$
\begin{aligned}
v^T L v &= \sum_{\{i,j\} \in E(G)} (v_i - v_j)^2 \\
&= \sum_{\{i,j\} \in E(G^-)} (v_i - v_j)^2 + \sum_{\{i,j\} \in E(G) \setminus E(G^-)} (v_i - v_j)^2 \\
&\geq \sum_{\{i,j\} \in E(G^-)} (v_i - v_j)^2 \\
&= v^T L^- v
\end{aligned}
$$

$\square$

**Lemma 2.** *Let $G$ and $G^-$ be two graphs with $V(G^-) = V(G)$, $E(G^-) \subseteq E(G)$. Let $L$ and $L^-$ be their respective Laplacians, with $k$th eigenvalue $\lambda_k$ and $\lambda_k^-$ respectively. Then $\lambda_k \geq \lambda_k^-$.*

*Proof.* From the min-max interpretation of eigenvectors, we have

$$
\lambda_k = \min_{S_k} \max_{v \in S_k} v^T L v
$$

where $S_k$ ranges over all $k$-dimensional subspaces. From lemma 1, we get

$$
v^T L^- v \leq v^T L V
$$

for all $v \in S_k$, and for all k-dimensional subspaces $S_k$. This gives us

$$
\max_{v \in S_k} v^T L^- v \leq \max_{v \in S_k} v^T L V
$$

for all k-dimensional subspaces $S_k$. Since this is true for all $S_k$, we get

$$
\min_{S_k} \max_{v \in S_k} v^T L^- v \leq \min_{S_k} \max_{v \in S_k} v^T L V
$$

From the min-max interpretation, we get $\lambda_k \geq \lambda_k^-$

$\square$

---

[*]Equal contribution

For all the proofs, unless otherwise stated, assume the first eigenvector (corresponding to the smallest eigenvalue) to be the all ones vector.

**Theorem 1.** *For all datasets $X$ and $X'$, such that $dist(X, X') \leq \varepsilon$, the function $F : \mathcal{X}_n \to \mathbb{R}^n$ obtained using the second eigenvector of the Laplacian as defined in the main paper satisfies*

$$\min(\|F(X) - F(X')\|, \|F(X) - (-F(X'))\|) \leq \left(2\sqrt{2}\right)\sqrt{\frac{\lambda_2^+(X) - \lambda_2^-(X)}{\lambda_3^-(X) - \lambda_2^-(X)}}.$$

*Proof.* For cleanliness of notation, let $G(X) = G$, $L(X) = L$, $\lambda_k(X) = \lambda_k$, $G^-(X) = G^-$, etc. Observe that

$$\lambda_2^+ \geq \lambda_2(X') = v_2(X')^T L(X') v_2(X') \geq v_2(X')^T L^- v_2(X')$$

The first part of the inequality follows from lemma 2, and second part follows from lemma 1. Now write

$$v_2(X') = \alpha v_2^- + \beta v_\perp$$

where $v_\perp$ is a unit vector perpendicular to $v_2^-$ and all-ones vector, and $\alpha$ and $\beta$ are scalars. Then

$$\begin{aligned}
\lambda_2^+ &\geq v_2(X') L^- v_2(X') \\
&= \left(\alpha v_2^- + \beta v_\perp\right)^T L^- \left(\alpha v_2^- + \beta v_\perp\right) \\
&= \alpha^2 v_2^{-T} L^- v_2^- + \beta^2 v_\perp^T L^- v_\perp^T \\
&\geq \alpha^2 \lambda_2^- + \beta^2 \lambda_3^-
\end{aligned}$$

using the property that the $v_2^-$ and $v_\perp$ are orthogonal, and $v_2^-$ is an eigenvector of $L^-$. By rearranging, and observing $\alpha^2 + \beta^2 = 1$ (as $v_2(X')$ is a unit vector), we get

$$\beta^2(\lambda_3^- - \lambda_2^-) \leq \lambda_2^+ - \alpha^2 \lambda_2^- - \beta^2 \lambda_2^- = \lambda_2^+ - \lambda_2^-$$

$$\beta^2 \leq \frac{\lambda_2^+ - \lambda_2^-}{\lambda_3^- - \lambda_2^-}$$

As $|\alpha| = \sqrt{1 - \beta^2}$, we get

$$\begin{aligned}
\min(\|v_2(X') - v_2^-\|_2^2, \|-v_2(X') - v_2^-\|_2^2) &= \min(2(1 - v_2(X')^T v_2^-), 2(1 + v_2(X')^T v_2^-)) \\
&= 2(1 - |\alpha|) \\
&= 2\left(1 - \sqrt{1 - \beta^2}\right) \\
&\leq 2\beta^2.
\end{aligned}$$

This gives us

$$\min(\|v_2(X') - v_2^-\|_2, \|-v_2(X') - v_2^-\|_2) \leq \sqrt{2}|\beta| \leq \sqrt{2}\sqrt{\frac{\lambda_2^+ - \lambda_2^-}{\lambda_3^- - \lambda_2^-}}$$

Notice that the above argument holds for any $X'$ such that $dist(X, X') \leq \varepsilon$; in particular, it holds for $X' = X$. This gives us

$$\min(\|v_2(X) - v_2^-\|_2, \|-v_2(X) - v_2^-\|_2) \leq \sqrt{2}\sqrt{\frac{\lambda_2^+ - \lambda_2^-}{\lambda_3^- - \lambda_2^-}}.$$

Using triangle inequality, we get

$$\min(\|v_2(X) - v_2(X')\|_2, \|v_2(X) - (-v_2(X'))\|_2) \leq 2\sqrt{2}\sqrt{\frac{\lambda_2^+ - \lambda_2^-}{\lambda_3^- - \lambda_2^-}}.$$

This finishes the proof as $F(X) = v_2(X)$ and $F(X') = v_2(X')$. $\square$

**Theorem 2.** *For any pair of datasets $X$ and $X'$, such that $dist(X, X') \leq \varepsilon$, there exists a $k \times k$ invertible matrix $M$, such that the features $F_X$ and $F_{X'}$ as defined in the main paper satisfy*

$$\sqrt{\sum_{i \in [n]} \|MF_X(x_i) - F_{X'}(x_i')\|^2} \leq \left(2\sqrt{2k}\right)\sqrt{\frac{\lambda_{k+1}^+(X) - \lambda_2^-(X)}{\lambda_{k+2}^-(X) - \lambda_2^-(X)}}$$

*Proof.* This proof generalizes the proof of theorem 1. For cleanliness of notation, let $G(X) = G$, $L(X) = L$, $\lambda_k(X) = \lambda_k$, $G^-(X) = G^-$, etc.

Let $v(X')$ be any unit vector in $S_k(X') = Span(v_2(X'), \ldots, v_{k+1}(X'))$. We will prove a bound on distance of $v(X')$ from its closest unit vector in $S_k^- = Span(v_2^-, \ldots, v_{k+1}^-)$. Write

$$v(X') = \sum_{i=2}^{k+1} \alpha_i v_i^- + \beta v_\perp$$

where $v_\perp$ is a unit vector satisfying $v_\perp \perp v_i^-(x)$ for all $1 \leq i \leq k+1$, and $\alpha_i$ and $\beta$ are scalars.

By lemma 2, we get

$$\lambda_{k+1}^+ \geq \lambda_{k+1}(X') = v_{k+1}(X')^T L(X') v_{k+1}(X')$$

and by lemma 1 and by the definition of eigenvectors, we get

$$v_{k+1}(X')^T L(X') v_{k+1}(X') \geq v(X')^T L(X') v(X') \geq v(X')^T L^- v(X')$$

The first part of the inequality follows from lemma 2, and second part follows from lemma 1.

Combining the two inequalities, we get

$$\lambda_{k+1}^+ \geq v(X')^T L^- v(X')$$

$$= \left(\sum_{i=2}^{k+1} \alpha_i v_i^- + \beta v_\perp\right)^T L^- \left(\sum_{i=2}^{k+1} \alpha_i v_i^- + \beta v_\perp\right)$$

$$= \sum_{i=2}^{k+1} \alpha_i^2 v_i^{-T} L^- v_i^- + \beta^2 v_\perp^T L^- v_\perp^T$$

$$\geq \sum_{i=2}^{k+1} \alpha_i^2 \lambda_i^- + \beta^2 \lambda_{k+2}^-$$

using the property that the $v_i^-$ and $v_\perp$ are all mutually orthogonal. Rearranging:

$$\beta^2(\lambda_{k+2}^- - \lambda_2^-) \leq \lambda_{k+1}^+ - \sum_{i=2}^{k+1} \alpha_i^2 \lambda_i^- - \beta^2 \lambda_2^-$$

$$\leq \lambda_{k+1}^+ - \lambda_2^-$$

which implies

$$\beta^2 \leq \frac{\lambda_{k+1}^+ - \lambda_2^-}{\lambda_{k+2}^- - \lambda_2^-}$$

For simplicity of notation, let

$$\alpha = (\alpha_2, \ldots, \alpha_{k+1}) \in \mathbb{R}^k, \qquad v^- = \frac{\sum_{i=2}^{k+1} \alpha_i v_i^-}{\|\alpha\|}$$

Then

$$\left\|v(X') - v^-\right\|_2^2 = 2\left(1 - v(X')^T v^-\right) = 2(1 - \|\alpha\|) = 2\left(1 - \sqrt{1 - \beta^2}\right) \leq 2\beta^2 \leq 2\frac{\lambda_{k+1}^+ - \lambda_2^-}{\lambda_{k+2}^- - \lambda_2^-}$$

This shows that every unit vector in $S_k(X')$ is within $\sqrt{2}\sqrt{\frac{\lambda_{k+1}^+ - \lambda_2^-}{\lambda_{k+2}^- - \lambda_2^-}}$ of some unit vector in $S_k^-$, and, by symmetry, vice-versa. Notice that the above argument holds for any $X'$ such that $dist(X, X') \leq \varepsilon$; in particular, it holds for $X' = X$. It thus follows from triangle inequality that every unit vector in $S_k(X')$ must be within $2\sqrt{2}\sqrt{\frac{\lambda_{k+1}^+ - \lambda_2^-}{\lambda_{k+2}^- - \lambda_2^-}}$ of some unit vector $S_k(X)$.

Let $F(X) = (F_X^1, F_X^2, \ldots, F_X^k)$ be an $n \times k$ matrix. Define $F(X')$ similarly. Observe that

$$\sqrt{\sum_{i \in [n]} \|MF_X(x_i) - F_{X'}(x_i')\|^2} = \|F(X)M^T - F(X')\|$$

Now, to prove the theorem we need to show the existence of an invertible matrix $M$ such that

$$\|F(X)M^T - F(X')\| \leq 2\sqrt{2k}\sqrt{\frac{\lambda_{k+1}^+ - \lambda_2^-}{\lambda_{k+2}^- - \lambda_2^-}}$$

When $2\sqrt{2}\sqrt{\frac{\lambda_{k+1}^+ - \lambda_2^-}{\lambda_{k+2}^- - \lambda_2^-}} \geq \sqrt{2}$, the desired bound is trivially true. To see this, set $M$ to be a diagonal matrix with diagonal entries $\pm 1$ such that $\langle M_{ii}F_X^i, F_{X'}^i \rangle \geq 0$ for all $i$. Since $F_X^i$ and $F_{X'}^i$ are unit vectors, we get

$$\|F(X)M^T - F(X')\| = \sqrt{\sum_{i \in [k]} \|M_{ii}F_X^i - F_{X'}^i\|^2}$$
$$\leq \sqrt{2k}$$
$$\leq 2\sqrt{2k}\sqrt{\frac{\lambda_{k+1}^+ - \lambda_2^-}{\lambda_{k+2}^- - \lambda_2^-}}$$

where $\|\cdot\|$ denotes the Frobenius norm for matrices, and $\ell_2$ norm for vectors.

So now assume $2\sqrt{2}\sqrt{\frac{\lambda_{k+1}^+ - \lambda_2^-}{\lambda_{k+2}^- - \lambda_2^-}} < \sqrt{2}$. Let $P$ be a projection map onto subspace $S_k(X)$. As $F_{X'}^1, \ldots, F_{X'}^k$ form an orthonormal basis for $S_k(X')$, projecting them onto $S_k(X)$ must yield a basis for $S_k(X)$; if not, then we would have $Pu = 0$ for some unit vector $u \in S_k(X')$, so that $u$ would be orthogonal (i.e. have distance $\sqrt{2}$) to every unit vector in $S_k(X)$, which contradicts the assumption that $2\sqrt{2}\sqrt{\frac{\lambda_{k+1}^+ - \lambda_2^-}{\lambda_{k+2}^- - \lambda_2^-}} < \sqrt{2}$. Now let $M \in \mathbb{R}^{k \times k}$ be an invertible matrix; such that $M^T$ corresponds to the change of basis matrix satisfying $PF(X') = F(X)M^T$ (this must exist since $PF(X')$ and $F(X)$ are both bases of $S_k(X)$).Then

$$\|F(X)M^T - F(X')\| = \|PF(X') - F(X')\| \leq 2\sqrt{2k}\sqrt{\frac{\lambda_{k+1}^+ - \lambda_2^-}{\lambda_{k+2}^- - \lambda_2^-}}$$

where last inequality follows since for each column vector of $F(X')$, its projection onto $S_k(X)$ has $\ell_2$ distance of at most $2\sqrt{2}\sqrt{\frac{\lambda_{k+1}^+ - \lambda_2^-}{\lambda_{k+2}^- - \lambda_2^-}}$ from it. This is because for each unit vector in $S_k(X')$, there is a vector in $S_k(X)$ at a distance of at most $2\sqrt{2}\sqrt{\frac{\lambda_{k+1}^+ - \lambda_2^-}{\lambda_{k+2}^- - \lambda_2^-}}$ from it. This concludes the proof. $\square$

## Proof of Theorem 3

**Theorem 3.** *For a sufficiently large training set size $n$, if $\mathbb{E}_{X \sim D}\left[(\lambda_3(X) - \lambda_2(X))^{-1}\right] \leq c$ for some small enough constant $c$, then with probability $0.95$ over the choice of $X$, the function $f_X$ :*

$\mathbb{R}^d \to \mathbb{R}$ *as defined above satisfies* $\Pr_{x \sim D}[\exists x' \text{ s.t. } dist(x, x') \leq \varepsilon \text{ and } |f_X(x) - f_X(x')| \geq \delta_x] \leq$
0.05, *for*

$$\delta_x = \left(6\sqrt{2}\right)\sqrt{\frac{\lambda_2^+(x) - \lambda_2^-(x)}{\lambda_3^-(x) - \lambda_2^-(x)}}.$$

*This also implies that with probability* 0.95 *over the choice of* X, $f_X$ *is* $(\epsilon, 20 \, \mathbb{E}_{x \sim D}[\delta_x], 0.1)$ *robust as per Definition 1.*

We give the proof of this theorem as a series of lemmas. First, similar to theorem 1, we show the robustness of function $f$ up to sign.

**Lemma 3.** *Given two points* $x$ *and* $x'$, *such that* $dist(x, x') \leq \varepsilon$, *the function* $f : \mathbb{R}^d \to \mathbb{R}$ *defined in the main text of the paper satisfies*

$$\min(|f(x) - f(x')|, |f(x) - (-f(x'))|) \leq \left(2\sqrt{2}\right)\sqrt{\frac{\lambda_2^+(x) - \lambda_2^-(x)}{\lambda_3^-(x) - \lambda_2^-(x)}}.$$

*Proof.* The proof follows from the same argument as Theorems 1 and 2 above. For completeness, we include it again here.

For cleanliness of notation, let $G(x) = G$, $L(x) = L$, $\lambda_k(x) = \lambda_k$, $G^-(x) = G^-$, etc. Observe that

$$\lambda_2^+ \geq \lambda_2(x') = v_2(x')^T L(x') v_2(x') \geq v_2(x')^T L^- v_2(x')$$

The first part of the inequality follows from lemma 2, and second part follows from lemma 1. Now write

$$v_2(x') = \alpha v_2^- + \beta v_\perp$$

where $v_\perp$ is a unit vector perpendicular to $v_2^-$ and all-ones vector, and $\alpha$ and $\beta$ are scalars. Then

$$\begin{aligned}
\lambda_2^+ &\geq v_2(x') L^- v_2(x') \\
&= \left(\alpha v_2^- + \beta v_\perp\right)^T L^- \left(\alpha v_2^- + \beta v_\perp\right) \\
&= \alpha^2 v_2^{-T} L^- v_2^- + \beta^2 v_\perp^T L^- v_\perp^T \\
&\geq \alpha^2 \lambda_2^- + \beta^2 \lambda_3^-
\end{aligned}$$

using the property that the $v_2^-$ and $v_\perp$ are orthogonal, and $v_2^-$ is an eigenvector of $L^-$. By rearranging, and observing $\alpha^2 + \beta^2 = 1$ (as $v_2(x')$ is a unit vector), we get

$$\beta^2(\lambda_3^- - \lambda_2^-) \leq \lambda_2^+ - \alpha^2 \lambda_2^- - \beta^2 \lambda_2^- = \lambda_2^+ - \lambda_2^-$$

$$\beta^2 \leq \frac{\lambda_2^+ - \lambda_2^-}{\lambda_3^- - \lambda_2^-}$$

As $|\alpha| = \sqrt{1 - \beta^2}$, we get

$$\begin{aligned}
\min(\left\|v_2(x') - v_2^-\right\|_2^2, \left\|-v_2(x') - v_2^-\right\|_2^2) &= \min(2(1 - v_2(x')^T v_2^-), 2(1 + v_2(x')^T v_2^-)) \\
&= 2(1 - |\alpha|) \\
&= 2\left(1 - \sqrt{1 - \beta^2}\right) \\
&\leq 2\beta^2.
\end{aligned}$$

This gives us

$$\min(\left\|v_2(x') - v_2^-\right\|_2, \left\|-v_2(x') - v_2^-\right\|_2) \leq \sqrt{2}|\beta| \leq \sqrt{2}\sqrt{\frac{\lambda_2^+ - \lambda_2^-}{\lambda_3^- - \lambda_2^-}}$$

Notice that the above argument holds for any $x'$ such that $dist(x, x') \leq \varepsilon$; in particular, it holds for $x' = x$. This gives us

$$\min(\left\|v_2(x) - v_2^-\right\|_2, \left\|-v_2(x) - v_2^-\right\|_2) \leq \sqrt{2}\sqrt{\frac{\lambda_2^+ - \lambda_2^-}{\lambda_3^- - \lambda_2^-}}.$$

Using triangle inequality, we get

$$\min(\|v_2(x) - v_2(x')\|_2, \|v_2(x) - (-v_2(x'))\|_2) \leq 2\sqrt{2} \sqrt{\frac{\lambda_2^+ - \lambda_2^-}{\lambda_3^- - \lambda_2^-}}.$$

Thus, we conclude

$$\min\left(|f(x) - f(x')|, |f(x) - (-f(x'))|\right)$$
$$\leq \min\left(\|v_2(x) - v_2(x')\|, \|v_2(x) - (-v_2(x'))\|\right)$$
$$\leq 2\sqrt{2} \sqrt{\frac{\lambda_2^+ - \lambda_2^-}{\lambda_3^- - \lambda_2^-}}$$

as desired. □

While flipping signs for the whole dataset is fine, we don't want the features of some of the points to flip signs arbitrarily. As described in the main text, to resolve this, we select the eigenvector $v_2(x)$ to be the eigenvector (with eigenvalue $\lambda_2(x)$) whose last $|X|$ entries have the maximum inner product with $v_2(X)$. We show a bound on this inner product next. Let $v_2^*(x)$ be a $|X|$ dimensional vector obtained by chopping off the first entry of $v_2(x)$.

**Lemma 4.** *For $v_2^*(x)$ defined as above, we have*

$$\langle v_2^*(x), v_2(X) \rangle \geq \sqrt{1 - \frac{\lambda_2(x) - \lambda_2(X)\left(1 - v_2(x)_0^2 \frac{n+1}{n}\right)}{\lambda_3(X) - \lambda_2(X)} - v_2(x)_0^2 \frac{n+1}{n}}$$

*Proof.* For cleanliness of notation, let $G(x) = G, L(x) = L, \lambda_k(x) = \lambda_k, v_2^*(x) = v_2^*$, etc. Write $v_2^* = \alpha v_2(X) + \beta w + \gamma \mathbf{1}/\sqrt{n}$ for scalars $\alpha, \beta, \gamma$ and vector $w$ orthogonal to $v_2(X)$ and $\mathbf{1}$. Taking inner product of both sides with $\mathbf{1}$, we get

$$\langle v_2^*, \mathbf{1} \rangle = \alpha \langle v_2(X), \mathbf{1} \rangle + \beta \langle w, \mathbf{1} \rangle + \gamma \langle \mathbf{1}/\sqrt{n}, \mathbf{1} \rangle$$

As $\langle v_2(X), \mathbf{1} \rangle = 0$, $\langle w, \mathbf{1} \rangle = 0$, and $\langle v_2^*, \mathbf{1} \rangle = -v_2(x)_0$, this gives us $\gamma = -v_2(x)_0/\sqrt{n}$.

Now, as $w$ is orthogonal to the bottom two eigenvectors of $L(X)$, we get $w^T L(X) w \geq \lambda_3(X)$. Then

$$\lambda_2 = v_2^T L v_2 \geq v_2^{*T} L(X) v_2^*$$
$$\geq \alpha^2 \lambda_2(X) + \beta^2 \lambda_3(X)$$
$$= \left(1 - \gamma^2 - v_2(x)_0^2\right)\lambda_2(X) + \beta^2(\lambda_3(X) - \lambda_2(X))$$

Putting $\gamma = -v_2(x)_0/\sqrt{n}$. and rearranging, we get

$$\beta^2 \leq \frac{\lambda_2 - \lambda_2(X)\left(1 - v_2(x)_0^2 \frac{n+1}{n}\right)}{\lambda_3(X) - \lambda_2(X)} \tag{1}$$

Notice now that $\alpha^2 + \beta^2 + \gamma^2 = \|v_2^*\|^2 = 1 - v_2(x)_0^2$. Since we know $\gamma^2 + v_2(x)_0^2 = v_2(x)_0^2 \frac{n+1}{n}$, and we know $\alpha \geq 0$ by definition, we conclude

$$\langle v_2^*(x), v_2(X) \rangle = \alpha = \sqrt{1 - \beta^2 - v_2(x)_0^2 \frac{n+1}{n}}$$

whereupon substituting the bound on $\beta^2$ from Eqn. (1) gives the desired result. □

Next, we give robustness bound on $f$ when $\langle v_2^*(x), v_2(X) \rangle > \frac{1}{\sqrt{2}}$

**Lemma 5.** *Given two points $x$ and $x'$, such that $dist(x, x') \leq \varepsilon$, if $\langle v_2^*(x), v_2(X) \rangle > \frac{1}{\sqrt{2}}$ the function $f_X : \mathbb{R}^d \to \mathbb{R}$ defined in the main text of the paper satisfies*

$$|f_X(x) - f_X(x')| \leq \left(6\sqrt{2}\right) \sqrt{\frac{\lambda_2^+(x) - \lambda_2^-(x)}{\lambda_3^-(x) - \lambda_2^-(x)}}.$$

*Proof.* If $\langle v_2(x), v_2(x') \rangle \geq 0$, then from the proof of Lemma 3, we get

$$
\begin{aligned}
|f_X(x) - f_X(x')| &\leq \|v_2(x) - v_2(x')\| \\
&= \min\left( \|v_2(x) - v_2(x')\|, \|v_2(x) - (-v_2(x'))\| \right) \\
&\leq 2\sqrt{2}\, \sqrt{\frac{\lambda_2^+ - \lambda_2^-}{\lambda_3^- - \lambda_2^-}}
\end{aligned}
$$

and we are done.

Otherwise, let $v$ be the vector $v_2(X)$ with a zero prepended to it. Note that $\langle v_2(x), v \rangle = \langle v_2^*(x), v_2(X) \rangle > \frac{1}{\sqrt{2}}$. Also, let $w$ be a unit vector in the direction of projection of $v$ on the subspace spanned by $v_2(x)$ and $v_2(x')$. And $w_\perp$ be a vector orthogonal to $w$ such that $v = \alpha w + \beta w_\perp$. This gives us $\langle v_2(x'), w \rangle = \frac{1}{\alpha} \langle v_2(x'), v \rangle \geq \langle v_2(x'), v \rangle \geq 0$ as $0 < \alpha < 1$. Similarly, $\langle v_2(x), w \rangle > \frac{1}{\sqrt{2}}$.

For three unit vectors $w, v_2(x), v_2(x')$ lying in a two dimensional subspace such that $\langle v_2(x), w \rangle > \frac{1}{\sqrt{2}}$ and $\langle v_2(x'), w \rangle \geq 0$, if $\langle v_2(x), v_2(x') \rangle < 0$, we get $\langle v_2(x), v_2(x') \rangle \geq \frac{-1}{\sqrt{2}}$. This implies $\langle v_2(x), -v_2(x') \rangle \leq \frac{1}{\sqrt{2}}$. From these inner product values, we get $\|v_2(x) - v_2(x')\| \leq 1.85$, and $\|v_2(x) - (-v_2(x'))\| \geq 0.75$.

This gives us

$$
\|v_2(x) - v_2(x')\| \leq 3 \min(\|v_2(x) - (-v_2(x'))\|, \|v_2(x) - v_2(x')\|) \leq 3 \cdot 2\sqrt{2}\, \sqrt{\frac{\lambda_2^+ - \lambda_2^-}{\lambda_3^- - \lambda_2^-}},
$$

where the second inequality now follows from the proof of Lemma 3.

As $|f_X(x) - f_X(x')| \leq \|v_2(x) - v_2(x')\|$, this finishes the proof. $\qquad\square$

Next, we show that under the conditions mentioned in our theorem, $\langle v_2^*(x), v_2(X) \rangle \geq \frac{1}{\sqrt{2}}$, for most $x \sim D$.

**Lemma 6.** *For a sufficiently large training set size $n$, if $\mathbb{E}_{X \sim D}\left[\frac{1}{\lambda_3(X) - \lambda_2(X)}\right] \leq c$ for some small enough constant $c$, then with probability $0.95$ over the choice of $X$,*

$$
\Pr_{x \sim D}\left[ \langle v_2^*(x), v_2(X) \rangle \geq \frac{1}{\sqrt{2}} \right] \geq 0.95
$$

*Proof.* From lemma 4, we know

$$
\langle v_2^*(x), v_2(X) \rangle \geq \sqrt{1 - \frac{\lambda_2(x) - \lambda_2(X)\left(1 - v_2(x)_0^2 \frac{n+1}{n}\right)}{\lambda_3(X) - \lambda_2(X)} - v_2(x)_0^2 \frac{n+1}{n}}
$$

For $n$ large enough, $\frac{n+1}{n} \approx 1$, so we need to show

$$
\frac{\lambda_2(x) - \lambda_2(X)\left(1 - v_2(x)_0^2\right)}{\lambda_3(X) - \lambda_2(X)} + v_2(x)_0^2 \leq \frac{1}{2}
$$

with probability $0.95$ over the choice of $X$, and $0.95$ over $x$.

By Markov's inequality, we get

$$
Pr_{X \sim D}[\lambda_3(X) - \lambda_2(X) \leq \frac{1}{100c}] \leq 0.01.
$$

For any size $n$ unit vector, if we pick one of its coordinates uniformly at random, it's expected squared value is $\frac{1}{n}$. By this argument, we get

$$
\mathbb{E}_{X \sim D, x \sim D}[v_2(x)_0^2] = \frac{1}{n+1}.
$$

By Markov's inequality, we get that

$$Pr_{X \sim D, x \sim D}[v_2(x)_0^2 \geq \frac{1000}{n+1}] \leq 0.001.$$

This implies that with probability $0.98$ over the choice of $X$,

$$Pr_{x \sim D}[v_2(x)_0^2 \geq \frac{1000}{n+1}] \leq 0.05.$$

Also, note that $\lambda_2(x) \leq \lambda_2(X) + 1$, since the second eigenvalue of a graph can go up by at most one by adding a new vertex. We also have $\lambda_2(X) \leq n$ since the eigenvalues of an un-normalized Laplacian are bounded by the number of vertices. Putting all this together, and applying union bound, we get that with probability $0.97$ over $X$, and $0.95$ over x,

$$\frac{\lambda_2(x) - \lambda_2(X)\left(1 - v_2(x)_0^2\right)}{\lambda_3(X) - \lambda_2(X)} + v_2(x)_0^2 \leq \frac{1 + n(\frac{1000}{n+1})}{\frac{1}{100c}} + \frac{1000}{n}$$

which is less than $\frac{1}{2}$ for small enough constant $c$, and $n$ large enough. $\qquad \square$

Combining Lemmas 5 and 6, we get that under the conditions stated in the theorem, with probability at least $0.95$ over the choice of $X$

$$\Pr_{x \sim D}\left[\exists x' \text{ s.t. } dist(x, x') \leq \varepsilon \text{ and } |f_X(x) - f_X(x')| \geq \delta_x\right] \leq 0.05$$

$$\text{for } \delta_x = \left(6\sqrt{2}\right)\sqrt{\frac{\lambda_2^+(x) - \lambda_2^-(x)}{\lambda_3^-(x) - \lambda_2^-(x)}}.$$

which finishes the proof. By applying Markov inequality and a union bound, we also get that $f_X$ as defined above is $(\epsilon, 20 \mathbb{E}_{x \sim D}[\delta_x], 0.1)$ robust with probability $0.95$ over the choice of $X$.

## Proof of Theorem 4

**Theorem 4.** *Assume that there exists some $(\varepsilon, \delta)$ robust function $F^*$ for the dataset $X$ (not necessarily constructed via the spectral approach). For any threshold $T$, let $G_T$ be the graph obtained on $X$ by thresholding at $T$. Let $d_T$ be the maximum degree of $G_T$. Then the feature $F$ returned by the spectral approach on the graph $G_{2\varepsilon/3}$ is at least $(\varepsilon/6, \delta')$ robust (up to sign), for*

$$\delta' = \delta\sqrt{\frac{8(d_\varepsilon + 1)}{\lambda_3(G_{\varepsilon/3}) - \lambda_2(G_{\varepsilon/3})}}.$$

*Proof.* Consider the graph $G_{2\varepsilon/3}$ obtained by setting the threshold $T = 2\varepsilon/3$ on the dataset $X$. Let $G_\varepsilon$ be the graph obtained by setting the threshold $T = \varepsilon$ on the dataset $X$, and let $G_{\varepsilon/3}$ be the graph obtained by setting the threshold $T = \varepsilon/3$ on the dataset $X$. Note that if all datapoints in $X$ are perturbed by at most $\varepsilon/6$ to get $X'$, then the inter point distances after perturbation are within $\varepsilon/3$ of the original inter point distances. Hence by Theorem 1,

$$\min(\|F(X) - F(X')\|, \|(-F(X)) - (F(X'))\|) \leq 2\sqrt{2}\sqrt{\frac{\lambda_2(G_\varepsilon) - \lambda_2(G_{\varepsilon/3})}{\lambda_3(G_{\varepsilon/3}) - \lambda_2(G_{\varepsilon/3})}}.$$

As $\lambda_2(G_\varepsilon) - \lambda_2(G_{\varepsilon/3}) \leq \lambda_2(G_\varepsilon)$, we will upper bound $\lambda_2(G_\varepsilon)$ to bound $\delta'$. Let $v$ be the vector such that the $i$th entry $v_i$ is $F_X^*(x_i)$, the feature assigned by $F^*$ to the datapoint $x_i$. Note that $\lambda_2(G_\varepsilon) \leq \sum_{(i,j) \in G_\varepsilon}(v_i - v_j)^2$, by using $v$ as a candidate eigenvector. We claim that $\lambda_2(G_\varepsilon) \leq (d_\varepsilon + 1)\delta^2$. To show this, we partition all edges in $G_\varepsilon$ into $t$ matchings $\{M_k, k \in [t]\}$. Note that for any matching $M_k$, $\sum_{(i,j) \in M_k}(v_i - v_j)^2 \leq \delta^2$. This follows by constructing the adversarial dataset $X'$ where each datapoint has been replaced by its matched vertex (if any) in the matching $M_k$, and by the fact that $F^*$ is $(\varepsilon, \delta)$ robust. By Vizing's Theorem, the number of matchings $t$ required is at most $d_\varepsilon + 1$. Therefore $\lambda_2(G_\varepsilon) \leq (d_\varepsilon + 1)\delta^2$ and the theorem follows. $\qquad \square$

## Proof of Theorem 5

**Theorem 5.** *Pick initial distance threshold $T = \sqrt{2} + 2\varepsilon$ in the $\ell_2$ norm, and use the first $N + 1$ eigenvectors as proposed in Section 2.1 of the main paper to construct a $N$-dimensional feature map $f : \mathbb{R}^d \to \mathbb{R}^N$. Then with probability at least $1 - N^2 e^{-\Omega(d)}$ over the random choice of training set, $f$ maps the entire inner sphere to the same point, and the entire outer sphere to some other point, except for a $\gamma$-fraction of both spheres, where $\gamma = Ne^{-\Omega(d)}$. In particular, $f$ is $(\varepsilon, 0, \gamma)$-robust.*

We will explain the construction of the features $f$ in detail at the end of the proof, as they become relevant. We give the proof of this theorem as a series of lemmas.

We use $S^{d-1}$ denote the unit sphere in $\mathbb{R}^d$ centered on the origin, and $rS^{d-1}$ denotes the sphere of radius $r$ centered on the origin. $\|\cdot\|$ denotes the $\ell_2$ norm.

**Lemma 7.** *Let $A$ be any point on $r_1 S^{d-1}$ and $B$ be a point chosen uniformly at random from $r_2 S^{d-1}$. Then the median distance $\|A - B\|$ is $M = \sqrt{r_1^2 + r_2^2}$. Further, for any fixed $\varepsilon > 0$, we have*

$$\Pr\left[\left|\|x - y\|^2 - (r_1^2 + r_2^2)\right| > \varepsilon\right] \leq \exp(-\Omega(d)) + \exp\left(-\Omega\left(\frac{\varepsilon^2 d}{r_1 r_2}\right)\right).$$

*Proof.* For notational simplicity let $A = r_1 x$ and $B = r_2 y$ for $x, y \in S^{d-1}$. Assume WLOG that $x = (1, 0, \ldots, 0)$. Suppose $y$ is drawn by drawing $z \sim N(0, I_d)$ and computing $y = z/\|z\|$. Then

$$\|A - B\|^2 = \|r_1 x - r_2 y\|^2 = r_1^2 + r_2^2 - 2r_1 r_2 y_1.$$

This immediately gives that the median value of $\|A - B\|$ is $M = \sqrt{r_1^2 + r_2^2}$. Further,

$$\left|\|x - y\|^2 - M^2\right| = 2r_1 r_2 |y_1| = 2r_1 r_2 \frac{|z_1|}{\|z\|}$$

Let $z = (z_1, z_2, \ldots, z_d)$, and consider $z' = (z_2, \ldots, z_d) \in \mathbb{R}^{d-1}$. Then $\|z'\|^2$, by definition, is a chi-square distributed random variable with $(d - 1)$ degrees of freedom that is independent of $z_1$. The following Chernoff bounds for chi-square variables applies:

$$\Pr\left[\|z'\|^2 < \frac{d - 1}{2}\right] \leq \left(\frac{e}{4}\right)^{(d-1)/4} = e^{-\Omega(d)}$$

$$\Pr\left[\|z'\|^2 > 2(d - 1)\right] \leq \left(\frac{e}{4}\right)^{(d-1)/4} = e^{-\Omega(d)}$$

Thus, for any fixed $\varepsilon, r_1, r_2$, we have

$$\begin{aligned}
\Pr\left[\left|\|x - y\|^2 - M^2\right| > \varepsilon\right] &= \Pr\left[2r_1 r_2 \frac{|z_1|}{\|z\|} > \varepsilon\right] \\
&\leq \Pr\left[|z_1| > \frac{\varepsilon \|z'\|}{2r_1 r_2}\right] \\
&\leq \Pr\left[\|z'\|^2 < \frac{d-1}{2}\right] + \Pr\left[|z_1| > \frac{\varepsilon \|z'\|}{2r_1 r_2} \,\Big|\, \|z'\|^2 \geq \frac{d-1}{2}\right] \\
&\leq e^{-\Omega(d)} + \Pr\left[z_1^2 > \frac{\varepsilon^2 (d-1)}{8 r_1 r_2}\right] \\
&\leq e^{-\Omega(d)} + \exp\left(-\Omega\left(\frac{\varepsilon^2 d}{r_1 r_2}\right)\right)
\end{aligned}$$

using independence of $z_1$ and $z'$, and Gaussian tail bounds on $z_1$. $\square$

Now let $x_1, \ldots, x_{2N}$ be a training set, where $x_1, \ldots, x_N$ are sampled i.i.d. uniform from $S^{d-1}$ and $x_{N+1}, \ldots, x_{2N}$ are sampled i.i.d. uniform from $RS^{d-1}$.

**Lemma 8.** *With probability at least $1 - O(N^2)e^{-\Omega(d)}$, both of these things hold:*

  *1. For every pair $(x_i, x_j)$ of points on the inner sphere, we have $\|x_i - x_j\| \leq \sqrt{2} + \varepsilon$*

2. *For every pair $(x_i, x_j)$ of points at least one of which is on the outer sphere, we have $\|x_i - x_j\| > \sqrt{2} + 3\varepsilon$*

*Proof.* By Lemma 7, for any constant $R > 1$ and $\varepsilon = (R-1)/8$, the probability that any given pair of points $x_i, x_j$ for $i < j \le 2N$ satisfies our condition is $1 - e^{-\Omega(d)}$ (since $\sqrt{2} + 3\varepsilon < \sqrt{1 + R^2} - \varepsilon$ for $\varepsilon = (R-1)/8$). Thus, the probability that all pairs satisfy our condition is $1 - O(N^2)e^{-\Omega(d)}$ as desired. ☐

Thus with probability $1 - O(N^2)e^{-\Omega(d)}$, the graph that we construct at distance threshold $T = \sqrt{2} + 2\varepsilon$ (with only the training points and no test point) has a very particular structure: one large connected component consisting of all the points on the inner sphere, and $N$ isolated points, one for each point on the outer sphere.

**Lemma 9.** *Let $x$ be a randomly drawn test point on the inner sphere. Then with probability at least $1 - O(N)e^{-\Omega(d)}$ over the choice of $x$, there is no $x'$ such that $\|x - x'\| \le \varepsilon$, and either (1) $\|x_i - x'\| > \sqrt{2} + 2\varepsilon$ for some $i \le N$, or (2) $\|x_i - x'\| \le \sqrt{2} + 2\varepsilon$ for some $i > N$.*

*Proof.* Certainly no such $i$ exists if $\|x_i - x\| \le \sqrt{2} + \varepsilon$ for all $i \le N$ and $\|x_i - x\| > \sqrt{2} + 3\varepsilon$ for all $i > N$. Using union bound and lemma 7, we get that a randomly drawn $x$ satisfies this with probability at least $1 - O(N)e^{-\Omega(d)}$. ☐

**Lemma 10.** *Let $x$ be a randomly drawn test point on the outer sphere. Then with probability at least $1 - O(N)e^{-\Omega(d)}$ over the choice of $x$, there is no $x'$ such that $\|x - x'\| \le \varepsilon$ and $\|x_i - x'\| \le \sqrt{2} + 2\varepsilon$ for any $i \le 2N$.*

*Proof.* Similar to the previous lemma, no such $i$ exists if $\|x_i - x\| > \sqrt{2} + 3\varepsilon$ for all $i \le 2N$. Using union bound and lemma 7, we get that a randomly drawn $x$ satisfies this with probability at least $1 - O(N)e^{-\Omega(d)}$. ☐

Lemmas 9 and 10 together give us the result that for almost all points, even after adversarial perturbation, the graph $G(x)$ we construct with threshold $T = \sqrt{2} + 2\varepsilon$ is identical for all points $x$ on the inner sphere, and identical for all points $x$ on the outer sphere (except a $\gamma = O(N)e^{-\Omega(d)}$ fraction): points on the inner sphere get connected to points on the inner sphere, and points on the outer sphere get connected to nothing. Since the map from graphs $G(x)$ to features $f(x)$ is deterministic, this means that $f$, in fact, maps all inner sphere points to one point and all outer sphere points to another point (except a $\gamma$-fraction); that is, $f$ is $(\varepsilon, 0, Ne^{-\Omega(d)})$-robust.

It only remains to show that $f$ maps inner-sphere and outer-sphere points to different outputs. Before proceeding further, we now fully explain the construction of the features $f$. Given a training set of $N$ points from the inner sphere and $N$ points from the outer sphere, construct the graph $G(X)$ and take the bottom-$(N + 1)$ eigenvectors $v_1, v_2, \ldots, v_{N+1}$ and prepend a 0 to each of them to yield $v'_1, v'_2, \ldots, v'_{N+1} \in \mathbb{R}^{2N+1}$. To compute the feature $f(x)$, we first construct the graph $G(x)$. Then, we project the vectors $v'_1, v'_2, \ldots, v'_{N+1}$ into the bottom-$(N + 1)$ eigenspace of $G(x)$ to yield vectors $u_1, u_2, \ldots, u_{N+1}$. The feature assigned to $G(x)$ is $f(x) := (u_{1,0}, u_{2,0}, \ldots, u_{N+1,0}) = e_0^T U \in \mathbb{R}^{N+1}$, where $U$ is the matrix whose columns are the $u_i$, and $e_0 = (1, 0, 0, \ldots, 0) \in \mathbb{R}^{2N+1}$. Similarly, define the matrix $V'$ corresponding to vectors $v'_i$. Let $P$ be the projection matrix onto the bottom-$(N + 1)$ eigenspace of $G(x)$

Assume WLOG that the first $N$ training examples are on the inner sphere, and the other $N$ are on the outer sphere. Then the vector

$$v^* := (0, \underbrace{1, 1, \ldots, 1}_{N}, \underbrace{0, 0, \ldots}_{N}, 0) \in \mathbb{R}^{2N+1}$$

is in the span of the $v'_i$, since it consists of a 0 prepended to a vector in the zero-eigenspace of $G(X)$. Suppose $v^* = \sum_i \alpha_i v'_i = V'\alpha$.

Notice that $e_0^T P V' \alpha = e_0^T P v^*$ will be 0 when $x$ is on the outer sphere, since $v^*$ is itself already in the zero-eigenspace of G(x) in this case. When $x$ is on the inner sphere, projecting $v^*$ onto

$G(x)$ will make its first component positive, since $v^*$ has positive dot product with the vector $u^* = (1, 1, 1, \ldots, 0, 0, \ldots)$ (which is an eigenvector of $G(x)$), and is orthogonal to every zero eigenvector of $G(x)$ orthogonal to $u^*$—and thus $e_0^T PV'\alpha > 0$ for $x$ on the inner sphere. Thus, $e_0^T U = e_0^T PV'$ must take different values on the outer and inner spheres. This concludes the proof of Theorem 5.