[Reviews · NeurIPS 2018]

Reviewer 1



The paper presents a view on adversarially robust features motivated by spectral graph theory. Bounds on the feature robustness are proved and shown to be empirically useful on toy datasets as well as a small-scale dataset (MNIST). The paper flows clearly and is well-motivated, and the connection to spectral graph theory is a very interesting non-obvious one, which gives the work novelty and originality. Some of the lower bounds are vacuous on realistic-scale problems, however the authors do well to clarify which theorems result in vacuous bounds and which do not. The theorems are also given along with toy examples which help readability. The main downside I found in this paper was the clarity of the writing itself, which can be improved. The paper includes quite a bit of informality and some typos, and some sentences which are unclear (e.g. line 37, inherent in what?). The related work also needs a revision: FGSM denotes an attack and not a defense (line 117, it seems that the paper is referring to adversarial training _using_ FGSM). Other papers that may help contextualize the work are [1] and [2], which also attempt to answer the "why" question of adversarial examples. The related work should also be its own section which contextualizes the work in the space of adversarial examples research, with background being reserved for defining adversarial examples, laplacians, etc. Specific/writing comments on intro: - Line 27: Should cite [3] which seems to be the first work on real-world examples - Line 31: Interest on -> interest in - Line 34: Models of -> Models for classifying - Line 35-36 are a bit unclear/hard to read [1] https://arxiv.org/abs/1804.11285 [2] https://arxiv.org/abs/1805.10204 [3] https://arxiv.org/abs/1707.07397

Reviewer 2



This paper focuses on adversarially robust machine learning. As existing literature struggles to develop adversarially robust models, the authors suggest to focus on building adversarially robust features. The authors present a method to build adversarially robust features, leveraging on the eigenvectors of the laplacian of a graph G obtained from the distances between the points in the training set. As a validation for their approach, the authors present a theoretical example where traditional methods (neural nets and nearest neighbors) fail to provide robust classifiers, while the proposed method provably provides robust features, and present experimental comparisons on MNIST data. Furthermore, the authors show that if there exists a robust function on the training data, then the spectral approach provides features whose robustness can be related to that of the robust function, which suggests that the spectral properties of the training data are related to the adversarial robustness. This intuition is also validated experimentally at the end of the paper. The paper reads well and the writing is insightful. The explanation of the method gives a clear intuition on why the proposed method makes sense to find adversarially robust features. Furthermore, I really appreciated the honesty of the authors in highlighting in several points the shortcomings of their approach. This paper does not solve the problem of adversarially robust learning, and the method may be computationally expensive. However, to the best of my knowledge, it is original and poses the basis of interesting future works. That said I have a concern: while the method to build adversarially robust features makes sense, it would be useful to better articulate why adversarially robust models/classifiers can be obtained using adversarially robust features. Is there an obvious implication stating that "if we use robust features, then we will get a robust model"? Are there cases where this implication fails? While this is is somewhat obvious for linear models (which explains the success of the experiments in sec. 5.1) I would like to see some theory - or at least some intuition - explaining why robust features make sense in general, also for non linear models, like neural nets.

Reviewer 3



The paper studies the very timely problem of feature robustness - it provides a new perspective on this problem with spectral approach. In particular, it wants to define functions that are robust to adversarial perturbations and vary across data points. Towards that goal, it shows connections between the spectral properties of the dataset, and the robustness of features. Detailed comments: - l.26: what are examples created in the real world? - l.48 why is it necessarily a good idea to disentangle the problems of accuracy, and robustness? any design that would try to address both simultaneously is necessarily bound to fail? - the notation for the distance metric, is the same as the dimension of the input space - this is not ideal. - l.79 - why does the variance have to be close to 1 necessarily? - l.127 - the statement that there is no effort in trying to understand the geometry properties in the robustness problem is probably an overstatement (see the work of Alhussein Fawzi for example, as mentioned later) - the section between lines 166 and 175, contains quite a few notation problems, or imprecisions; it might be worth revising that part carefully. Also, it it not clear at this stage why the dataset is augmented with x? And does it mean that a new graph is built for each new x? - at several places, the authors talk about suitable conditions for arguments or theorems (like in line 200 for example). These conditions should be clarified in the text - it would be a nice addition to have a sketch of the Th.2 in the main text. And actually, the discussion of Th.1 should come before Th 2 for better readability. - is there any constructive way to pick the correct threshold in lines 226-228? - the spectral view might appear as an overstatement in general - the authors mostly look at the Fiedler vector, and not at the full spectrum, in their analysis, and fonction construction. - the bounds defined by the authors, appear to be very loose, mostly because of the underlying assumptions: are they useful at all at the end? The authors should discuss in more details the relevance of the proposed bounds (esp. the lower one). - Figure 2: this one is not very convincing: one can argue against the author’s statement that there is a clear (linear) relationship + outliers... - recent works have studied the bounds on robustness, and relations to accuracy, e.g., the 2018 arxiv paper of Fawzi et al.. These works have to be discussed to provide a complete overview of the closely related literature, as they address some aspects of the key questions raised at l.38. - the are a few English typos, that should be corrected. E.g., lines 10,18, 133, 167 (index 0 misplaced), 169 (0 should be x_0??), 213, 336... Overall, the paper takes an interesting, and new spectral perspective on robustness. They derive also new bounds, that could potentially bring interesting insights on the robustness problem. The classification function stays unfortunately trivial (and classical in graph-based problem), and is not learned or trained for the target task, as this was hinted in the introduction. As a result, the paper brings interesting insights, but fails to be completely convincing about the relevance of this theoretical study and its results. It could be accepted however, as it will probably trigger interesting discussions on a timely topic.